# Extra- and Intra-Cellular Mechanisms of Hepatic Stellate Cell Activation

**DOI:** 10.3390/biomedicines9081014

**Published:** 2021-08-14

**Authors:** Yufei Yan, Jiefei Zeng, Linhao Xing, Changyong Li

**Affiliations:** Department of Physiology, School of Basic Medical Sciences, Wuhan University, Wuhan 430017, China; yanyufei@whu.edu.cn (Y.Y.); 2019305231065@whu.edu.cn (J.Z.); 2019305231078@whu.edu.cn (L.X.)

**Keywords:** hepatic fibrosis, hepatic stellate cell, myofibroblast, signal pathway

## Abstract

Hepatic fibrosis is characterized by the pathological accumulation of extracellular matrix (ECM) in the liver resulting from the persistent liver injury and wound-healing reaction induced by various insults. Although hepatic fibrosis is considered reversible after eliminating the cause of injury, chronic injury left unchecked can progress to cirrhosis and liver cancer. A better understanding of the cellular and molecular mechanisms controlling the fibrotic response is needed to develop novel clinical strategies. It is well documented that activated hepatic stellate cells (HSCs) is the most principal cellular players promoting synthesis and deposition of ECM components. In the current review, we discuss pathways of HSC activation, emphasizing emerging extra- and intra-cellular signals that drive this important cellular response to hepatic fibrosis. A number of cell types and external stimuli converge upon HSCs to promote their activation, including hepatocytes, liver sinusoidal endothelial cells, macrophages, cytokines, altered ECM, hepatitis viral infection, enteric dysbiosis, lipid metabolism disorder, exosomes, microRNAs, alcohol, drugs and parasites. We also discuss the emerging signaling pathways and intracellular events that individually or synergistically drive HSC activation, including TGFβ/Smad, Notch, Wnt/β-catenin, Hedgehog and Hippo signaling pathways. These findings will provide novel potential therapeutic targets to arrest or reverse fibrosis and cirrhosis.

## 1. Introduction

Liver fibrosis is the pathologic sequela of chronic repetitive injury and is a reversible healing response in response to acute or chronic cell injury. Further development of liver fibrosis leads to cirrhosis and even liver cancer. Various factors can cause liver fibrosis, the main risk factors identified at present include viral infection, alcoholism, obesity-related steatohepatitis and so on [1,2,3]. Cirrhosis is a major cause of morbidity and mortality globally, imposing a heavy health burden on many countries. Globally, cirrhosis currently causes 1.16 million deaths and is the 11th most common cause of death each year [4]. Cirrhosis imposes a huge economic burden in the United States, with estimated annual direct costs of more than USD 2 billion and indirect costs of more than USD 10 billion [5].

Hepatic stellate cell (HSC) activation represents a critical event in fibrosis [6,7]. In normal liver, HSCs exist in a quiescent non-proliferative state, having a star-like shape with intracellular lipid droplet storage containing vitamin A as retinyl palmitate [8]. HSCs are a type of resident non-mesenchymal cells that have features of both resident fibroblasts (embedded in normal stroma) and pericytes (endothelial cells attached to capillaries). Locating in the space of Disse, HSCs are a major producer of extracellular matrix (ECM) [8,9,10], which accounts for approximately 15% of total resident cells and one third of the total nonparenchymal cells in the normal human liver [11]. Pathological, toxic, metabolic or viral diseases lead to liver cell damage and immune cell infiltration, activating the transdifferentiation of HSCs to myofibroblasts, which is known as “activation”. It is generally believed that HSCs are the main source of myofibroblasts during hepatic fibrosis and are independent of the source of damage [12,13]. In chronic liver disease, the imbalance between the pro-fibrogenic and anti-fibrogenic mechanisms leads to continuous activation of proliferating, contractile, and migrating myofibroblasts, resulting in excessive production of ECM. Large amounts of ECM deposited in the liver lead to liver fibrosis [14]. Here, we review extra- and intra-cellular mechanisms of HSC activation, emphasizing recent emerging cellular and molecular signals that trigger this important cellular response to liver injury.

## 2. Extracellular Factors of HSC Activation

The extracellular factors that promote HSC activation have been identified as stimulation of various cell types, altered extracellular matrix, enteric dysbiosis, chronic infection of hepatitis virus, lipid metabolism disorder, exosomes, microRNA and other factors including alcohol, drugs and parasites (Figure 1).

### 2.1. Hepatocytes

In response to injury, hepatocytes change their gene expression and secretion profile, and thus affect HSC activation. Damage-associated molecular patterns (DAMPs) released by injured hepatocytes might directly or indirectly promote HSC activation. Nucleotide binding oligomerization domain-like receptors 3 (NLRP3) is one of the main components of inflammasomes and the downstream targets of DAMPs. Mice with the constitutively active mutant *NLRP3* develop severe liver inflammation with pyroptotic hepatocyte death and HSC activation [15]. When the liver is damaged, hepatocytes release IL-33, which activates the innate lymphoid cells (ILCs). In the three known cell subsets of ILCs (ILC1, ILC2 and ILC3), ILC2 drives HSC activation and promotes the occurrence of liver fibrosis, demonstrating that hepatocytes promote HSC activation [16]. In addition, damaged hepatocytes, rather than normal hepatocytes, secrete exosomes which contain microRNAs that activate HSCs [17].

### 2.2. Liver Sinusoidal Endothelial Cells

In the normal liver, liver sinusoidal endothelial cells (LSECs) maintain the quiescence of HSCs through heparin-binding EGF-like growth factor and paracrine factors such as nitric oxide (NO) [18,19]. Normal LSECs are fully differentiated and highly endocytic, which contain fenestrae. Prior to fibrosis, LSECs lose their fenestration and undergo capillarization due to incomplete differentiation of bone marrow-derived LSECs that are recruited to the injured liver, and are permissive for HSC activation [19,20,21]. In rats with thioacetamide-induced cirrhosis, a soluble guanylate cyclase activator, BAY 60-2770, leads to reversal of the capillarization, which further leads to quiescence of HSC and regression of fibrosis [21]. Depending on the injury environment, LSECs may promote either liver regeneration or fibrosis. In particular, the CXCR7–ID1 pathway in LSECs in response to injury promotes liver regeneration, but the FGFR1–CXCR4 pathway promotes HSC activation and fibrosis [22].

### 2.3. Macrophages

Accumulating evidence suggests that progressive fibrotic diseases, including hepatic fibrosis, are tightly regulated by macrophages [23]. Macrophages play dual roles in liver fibrosis progression and its resolution. Polarized and plastic activation of macrophages is traditionally classified into classic M1 and alternative M2 activation. The M1 phenotype is characterized by high expression of pro-inflammatory cytokines, high production of reactive nitrogen and oxygen intermediates, promotion of Th1 response, while the M2 phenotype characterized by highly efficient phagocytic activity, high expression of scavenging, mannose and galactose receptors and production of ornithine and polyamines via the arginase pathway [24]. During the progression of fibrosis, injury-induced inflammation triggers the recruitment of macrophages to the liver, where they produce cytokines and chemokines including transforming growth factor β (TGFβ), Platelet derived growth factor (PDGF), tumor necrosis factor (TNF), IL-1β, monocyte chemotactic protein-1 (MCP1), CCL3 and CCL5 to induce HSC activation. The recruitment of immature monocyte-derived LY6C^hi^ macrophages are facilitated by CCL2 secreted by Kupffer cells and HSCs. However, during the regression of liver fibrosis, macrophages have a CD11b^hi^/F4/80^int^LY6C^low^ phenotype which is arisen from a phenotypic switch of profibrogenic LY6C^hi^ macrophages. The LY6C^low^ phenotype stops the production of fibrogenic and inflammatory factors, and instead secrete matrix metalloproteinases (MMPs) such as MMP9 that promotes HSC apoptosis and MMP12 [18,25].

### 2.4. Fibrogenic Cytokines

TGFβ is generally considered to be the most impotent fibrogenic cytokine, as described in more detail in the following article. The downstream connective tissue growth factor (CTGF) of TGFβ is a key fibrogenic cytokine that accelerates the activation of HSCs. It has been reported that TGFβ induces the expression of CTGF through Smad and Stat3 signaling pathways in HSCs. In A-HSCs, the pro-fibrotic CTGF is also upregulated and promotes the pathogenetic processes of hepatic fibrosis, including cell proliferation, contractility, migration and ECM production [26]. A-HSCs release CTGF and other pro-fibrogenic factors which drive the deposition of ECM [27].

Interleukin plays an important role in the activation of hepatic stellate cells. interleukin-13 (IL-13) is an immunoregulatory cytokine secreted mainly by a T-cell subset termed Th2 cells. In HSCs, IL-13 directly induces the expression of collagen I and other key fibrosis-related genes such as α-smooth muscle actin (*α-SMA*) [28,29]. IL-13 also induces CTGF through the Erk-mitogen-activated protein kinase (MAPK) pathway to accelerate the activation of HSCs [30]. Damaged hepatocytes secreted IL-33 which leads to accumulation and activation of innate lymphoid cells (ILC2). ILC2 Activated by IL-33 produce IL-13, inducing the activation and trans-differentiation of HSCs [16,28]. IL-17 is produced mainly by Th17 cells, but can also be produced by neutrophils and other lymphocytes. IL-17 induces the production of collagen I in HSCs by activating the STAT3 signaling pathway and pharmacological inhibition of IL-17-induced ERK1/2 or p38 significantly reduces HSCs activation and collagen expression [28].

Dead or dying endothelial cells and white blood cells release inflammatory mediators, DAMPs or danger signals, which initiate a noninfectious “sterile” inflammatory response. Among them, TNF, IL-6, IL-1β, reactive oxygen species (ROS), and Hedgehog (Hh) ligand can facilitate the initiation process of HSC activation [31,32]. TNF and IL-1β cannot directly promote HSC activation, but they prolong the survival of A-HSCs through activating NF-κB signaling pathway both in vivo and in vitro [33]. ROS provides paracrine activation signals to HSCs. When the transmembrane enzyme complex Nox1 or Nox4 regulating ROS is inactivated, liver injury, inflammation and fibrosis were significantly reduced [34].

Platelets are also important cells involved in inflammation, and PDGF and TGFβ produced by them are important cytokines that induce HSC activation [35]. PDGF is an important mitogen in the liver and one of the chemokines that promote the proliferation and migration of HSCs. Studies in humans and rodents have shown that PDGF ligands and receptors are rapidly expressed in HSCs at the onset of liver injury.

### 2.5. Altered ECM

A-HSCs are a major producer of ECM, and the alteration of ECM also affects HSC activation. In normal liver, laminins, type IV collagen and a mixture of proteoglycans are scattered within the hepatic ECM. HSCs express two types of collagen receptors: integrins and discoidin domain-containing receptors, and each type receives signals from ECM components to regulate cell adhesion, differentiation, proliferation and migration [25]. HSCs secrete a large amount of ECM after activation, and then progressive deposition of ECM proteins in the space of Disse gradually leads to increased density and stiffness of ECM. Furthermore, matrix composition shifts from collagen type IV, heparan sulfate proteoglycan, and laminin to fibrillar collagen type I and III. These changes act as mechanical stimuli to activate HSCs at least partially through integrin signaling pathways, forming positive feedback loops [31]. In addition, the expanded ECM promotes proliferation of HSCs by binding PDGF, hepatocyte growth factor (HGF), fibroblast growth factor (FGF), epidermal growth factor (EGF) and vascular endothelial growth factor (VEGF) and other growth factors to play a storage role [36]. Furthermore, extracellular matrix protein 1 (ECM1) produced by hepatocytes interacts with αv integrins to stabilize extracellular matrix-deposited TGFβ to prevent HSC activation [37].

### 2.6. Enteric Dysbiosis

Gut microbes have many physiological functions, such as producing vitamin B series, digesting food particles and gaining energy from them, promoting the host immunity and antagonizing foreign invaders. Moreover, gut microbes can affect the normal physiological function of the liver through enterohepatic circulation. Normally, gut microbes play a protective role in the liver. For instance, commensal microbiota has a hepatoprotective effect, preventing liver fibrosis by reducing HSC activation [38]. However, enteric dysbiosis can cause pathological changes in the liver through HSC activation. Intestinal dysbiosis leads to release and increased exposure to pathogen-associated molecular patterns (PAMPs), activating HSCs through Toll-like receptors (TLRs) [39]. A high-fat diet increased the rate of endotoxin/lipopolysaccharide production by intestinal Gram-negative bacteria, leading to higher bacterial translocation rate, and accelerated fibrous formation in CCl_4_ and bile duct ligation (BDL) mice by promoting HSC activation [40].

### 2.7. Chronic Infection of Hepatitis Virus

Chronic infection of hepatitis virus has become one of the major risk factors for liver fibrosis worldwide [41]. Viral genes and proteins directly or indirectly promote HSC activation.

Hepatitis B virus (HBV) e antigen directly induced expression of TGFβ, and TGFβ in turn mediated the activation and proliferation of HSCs. Meanwhile, HBV e antigen promotes the release of soluble mediators that activate HSCs, resulting in the production of ECM components and related factors that lead to fibrogenesis in patients with chronic HBV infection [42]. HBV Dane particles and x and c proteins may up-regulate the mRNA levels of *PDGFβ* and *PDGFR-β* and promote the phosphorylation of PDGFR-β, leading to subsequent auto-phosphorylation. Furthermore, which induces HSC proliferation [43].

Hepatitis C virus (HCV) infects about 120–130 million people around the world. Chronic HCV infection is the cause of hepatic necroinflammatory lesions and fibrosis of variable intensity. HCV cannot directly infect human HSCs, but it has been proved that HCV viral proteins activate HSCs after direct interaction with plasma membrane in a variety of in vitro experiments [44]. Elevated expression of IL-34 and macrophage colony-stimulating factor (M-CSF) in HCV-infected hepatocytes stimulates the process of peripheral blood mononuclear cells transforming into macrophages and promotes HSC activation by enhancing TGFβ and PDGFβ signaling [45].

### 2.8. Lipid Metabolism Disorder

The retinoid in the human body is mainly stored in the lipid droplets of HSC cytoplasm. Under normal conditions, Q-HSCs store up to 80% of body retinols (vitamin A lipid droplets) and contribute to retinol homeostasis with visible lipid droplets in the cytoplasm. Alcohol dehydrogenases (ADHs) in HSCs are a kind of retinol metabolic enzyme that oxidizes retinol to retinaldehyde. Among the 6 different types in the ADH family, ADH3 promotes HSC activation and inhibits the activity of NK cells, which plays an important role in promoting the progression of liver fibrosis. ADH3 inhibition enhances cytotoxicity of NK cells against HSCs and reduces the expression of TGFβ1 and collagen. Ablation of *ADH3* gene blocked retinol metabolism in HSCs, alleviating liver fibrosis induced by BDL and CCl_4_ [46].

The involvement of cholesterol metabolism in HSC activation is not well understood, but disorders of cholesterol metabolism in other liver resident cells types may indirectly lead to HSC activation [47]. Alterations in cholesterol metabolism in nonalcoholic fatty liver disease (NAFLD) can activate Kupffer cells and induce HSC transdifferentiation [48]. Liver x receptors (LXRs) are the key regulator of cholesterol balance that govern whole body cholesterol homeostasis. Primary Lxrαβ^−/−^ HSCs are pro-fibrotic and pro-inflammatory. These cells lose their lipid droplets more rapidly during in vitro activation and achieve the activated phenotype more quickly than cells isolated from wild-type mice [49]. Acyl-coenzyme A: cholesterol acyltransferase (ACAT1) catalyzes the conversion of free cholesterol into cholesterol ester, which avoids the excessive accumulation of free cholesterol. ACAT1 deficiency leads to elevated free cholesterol levels in HSCs, enhanced TLR4 signaling and down-regulated bone morphogenetic protein and activin membrane-bound inhibitor expression, leading to HSC sensitivity to TGFβ activation [50]. Increased production of TGFβ and other potentially unknown signaling molecules by hepatocytes induced HSC activation even in the absence of immune cells due to excessive lipid accumulation in hepatocytes (e.g., during hepatic steatosis) [47]. Treatment of hepatocytes with palmitic acid not only induced hepatocyte apoptosis, but also enhanced the ability of hepatocyte-derived exosomes to activate HSCs [17].

### 2.9. Exosome and MicroRNA

As important means of communication between cell populations, exosomes are nano-sized membrane vesicles that can transfer lipid, nucleic acids, proteins, and other bioactive molecules between different cell populations. Exosomes can be released by various cells, and exert numerous physiological and pathological activities, including cell growth, proliferation, differentiation, and apoptosis [51]. Recently, various cell types in the liver including hepatocytes and LSECs have been shown to interact with HSCs via exosomes, in turn modulating the biological activities of HSCs. For instance, palmitic acid stimulation enhanced the production of exosomes in hepatocytes and changed their exosomal miRNA profile. Moreover, exosomes derived from these hepatocytes stimulated the activation of HSCs [17]. LSECs secrete exosomes that express high amounts of sphingosine kinase 1, which promote HSC migration and activation [52].

A variety of microRNAs have been reported to have the potential to regulate fibrogenic signaling pathways in HSCs and participate in the activation process of HSCs, including TGF-β/Smad, Wnt/β-catenin, Hedgehog and so on [53]. For instance, miR-214 is significantly upregulated during HSC activation and leads to ECM accumulation by inhibiting the expression of suppressor-of-fused homolog, a negative regulator of hedgehog signaling pathway in LX-2 cells [54]. MiR-125b can promote HSC activation and fibrogenesis by upregulating RhoA signaling pathway and can be considered as A-HSCs specific fibrosis marker [55]. MiR-195 overexpression activates HSCs by reducing Smad7, and its inhibitors block HSC activation, reduce *α-SMA* expression, and enhance *Smad7* expression [56]. These above observations suggest that exosomes and microRNAs may provide new clues for the therapeutic and diagnosis of hepatic fibrosis in the near future.

### 2.10. Other Factors

Other factors including alcohol, drugs and parasites can also influence HSC activation.

Studies have identified the impact of alcohol on the expression of epigenetic regulators during HSC activation. Alcohol directly affects HSC activation by stimulating overall changes in chromatin structure, leading to increased expression of ECM proteins. Furthermore, alcohol has the potential to promote the accumulation of elastin through directly stimulating tropoelastin gene transcription, elastin protein expression and *TIMP-1* gene transcription in HSCs [31,57]. In addition, alcohol inhibits the antifibrotic process by inhibiting natural killer cell-mediated interferon-gamma-induced death of A-HSCs [58].

Some drugs affect HSC activation. Methotrexate (MTX) is commonly used for the treatment of autoimmune diseases and skin diseases, but rheumatoid arthritis and psoriasis patients who receive MTX therapy for a long time are at high risk of developing liver damage. MTX-PG, a metabolite of MTX, inhibits 5-aminoimidazole-4-carboxamide ribonucleotide transformylase enzyme, leading to intracellular adenosine accumulation, which in turn leads to HSC activation, ECM accumulation and liver fibrosis [59]. Acetaminophen (APAP) is one of the quantitively most consumed drugs worldwide, but overdosing often results in severe liver damage and even liver failure [60]. APAP exposure does not directly cause HSC activation, but leads to toxicity mainly in hepatocytes and mounts a hepatocyte damage dependent activation of HSCs [61,62].

It has been reported that there are complex and diverse interactions between HSCs and *schistosome* eggs. The number of A-HSCs increased in murine and human livers infected with *schistosoma mansoni* compared with healthy liver [63]. It is possible that one role for A-HSCs is to coordinate the influx of the various immune cells to mediate the granulomatous response [64]. In addition, during *schistosome* infection, hepatocytes overexpress IL-33, driving the activation and proliferation of a subset of hepatic innate lymphoid cells (ILC2). In turn, ILC2s produce IL-13, which drives HSC activation by regulating *TGF-β1* and *CTGF* expression [65].

## 3. Intracellular Signaling Pathways of HSC Activation

The activation of HSCs is related to a variety of cytokines and constitutes a complex regulatory network, and some of the important signal transduction pathways have gradually become clear. Herein we describe the range of intracellular signaling pathways that individually or collectively drive HSC activation (Figure 2).

### 3.1. TGF-β/SMAD Pathway

TGF-β signaling is considered the key fibrogenic pathway that drives HSC activation and induces ECM production. In normal liver, Q-HSCs express trace amounts of TGF-β, which is up-regulated shortly after liver injury [66]. Active HSCs produce TGF-β in response to liver injury, which forms a positive feedback loop driving fibrogenesis through SMAD2/SMAD3, while SMAD7 inhibits the activation [67]. In Q-HSCs, TLR4 activation down-regulates the TGF-β pseudo receptor Bambi to stimulate HSCs to TGF-β-induced signals [68]. Apoptotic body-engulfing macrophages secrete TGF-β and activate HSCs [69,70]. ECM1, mainly produced by hepatocytes, attenuates activation of TGF-β and its activation of HSCs to prevent liver fibrosis [37]. TGFβ-1-induced transcript 1 protein (TGFβ1i1), also named as hydrogen peroxide-inducible clone-5 (Hic-5), inhibits the activation of HSCs and liver fibrosis through reducing the TGF-β/Smad2 signaling by upregulation of Smad7 [71]. The following molecules interact with TGF-β signaling and contribute to HSC activation.

Hyaluronan (HA) is a major extracellular matrix glycosaminoglycan and a biomarker for cirrhosis. The production and deposition of HA replace functional liver tissues feature prominently in liver fibrosis. *HA* and HA synthase 2 (*HAS2*) expression was elevated in both human and murine liver fibrosis. *HAS2* was transcriptionally up-regulated by TGF-β via Wilms tumor 1 to promote fibrogenic, proliferative, and mediates HSC activation through CD44, TLR4, and Notch1. Furthermore, *HA* expression and liver fibrosis were reduced upon *HAS2* inhibition and enhanced upon *HAS2* overexpression in HSCs. Depletion of HA synthesis by 4-methylumbelliferone suppresses HSC activation and liver fibrosis in mice, which may have potential to be a new therapeutic route for liver fibrosis [72].

Galectins are a family of animal beta-galactoside-binding lectins [73]. Galectin-3 has been implicated in a variety of biological processes including cell proliferation, adhesion, survival, and in the development of acute inflammation [74]. Disruption of the *Gal-3* gene blocks HSC activation and collagen expression, thus reducing liver fibrosis. Specifically, in CCl_4_-treated, *Gal-3*-deficient mice, HSC activation and collagen deposition are suppressed compared with wild-type animals [75]. Additionally, treatment of LX-2 cells with recombinant Gal-1 protein can increase the phosphorylation of SMAD2, SMAD3 and ERK1/2, and bind to neuronilin-1 in a glycosylation-dependent manner to enhance HSC migration [76].

HAb18G/CD147, a tumor-related glycoprotein expressed on the cellular membrane of HSCs, is highly expressed on activated HSCs. TGF-β upregulated *HAb18G/CD147* expression in LX-2 cells, and *HAb18G/CD147* transfection enhanced the profibrogenic genes expression. In mouse liver fibrosis model, *HAb18G/CD147* expression increased upon the development of fibrogenesis and decreased during the liver fibrosis recovery. These data implicate that HAb18G/CD147 plays a role in HSC activation and is an effective therapeutic target in fibrosis [77].

Bone morphogenetic proteins (BMPs) are members of the TGF-β superfamily and have effects on liver fibrosis development and progression, which play essential roles during embryonic development [78]. In the process of CCl_4_-induced liver fibrosis in mice, the expression of BMP7 increased first and then decreased as well as in human patients with CLD [79]. The results of in vitro experiments show that high doses of exogenous BMP7 can inhibit the activation, migration and proliferation of TGF-β1 induced HSCs. This effect is related to the up-regulation of pSMAD1/5/8 and down-regulation of the phosphorylation of SMAD3 and p38 by BMP7. Thus, exogenous BMP7 may be used as an anti-liver fibrosis drug [80]. Intriguingly, BMP6 is upregulated in NAFLD but not in other mouse liver injury models or diseased human livers (ALD and chronic HBV or HCV infection). Recombinant BMP6 suppresses the activation of HSCs and reduces proinflammatory and profibrogenic gene expression in activated HSCs [81].

### 3.2. Notch Signaling

The Notch signaling pathway enables cells to communicate with their direct neighbors by ligand–receptor interaction to convey the signal into a transcriptional response to regulate tissue and organ development [82]. In mammals, there are four known receptors (Notch 1–4) and five ligands belonging to the Jagged (Jagged1, 2) and Delta-like (Delta-like, Dll1, 3, and 4) family. Signaling upon ligand–receptor binding leads to sequential proteolytic cleavage processes in the Notch receptor extracellular and transmembrane domain to release the Notch intracellular domain (NICD). In the nucleus, NICD binds to the DNA-binding recombination signal binding protein (RBP)-Jκ and activates the transcription of target genes *Hes1*, *Hey1* and *Hey2* [83].

Rat HSCs express Notch receptors in vitro and up-regulate *JAG1* upon activation and differentiation to myofibroblasts [84]. The role of JAG1 in HSC biology is elusive, and more recent studies show that exposure of HSCs to JAG1 promotes α-SMA and collagen expression [84]. In TGF-β-induced human HSCs, fibrosis-related genes (*col I* and *α-SMA*), and *Notch3*, *JAG1* and *Hes1* were overexpressed compared to non-activated cells [85]. While Notch signaling can directly activates HSCs, Notch activation in neighboring cells (e.g., LSECs) also leads to HSC activation and the subsequent hepatic fibrosis. Notch activation down—regulates eNOS—sGC signaling, resulting in increased LSEC dedifferentiation, HSC activation and fibrosis [86]. Moreover, hepatocyte-specific Notch depletion in NASH mice leads to reduced fibrotic deposition and HSC activation [87], and hepatocyte Notch activation is sufficient to induce β-catenin-inactive HCC in mice with NASH [88]. Recently, a nanoparticle-mediated delivery system to target *γ*-secretase inhibitor to liver (GSI NPs) reduced liver fibrosis and inflammation in mice fed a NASH-provoking diet, without apparent gastrointestinal toxicity [89].

### 3.3. Wnt/β-Catenin Signaling

The Wnt pathway is commonly divided into β-catenin dependent (canonical) and independent (non-canonical) signaling [90]. In canonical Wnt signaling, (i) the extracellular Wnt protein is connected to the frizzled protein (Frz) on the target cell membrane and the co-receptor low-density lipoprotein receptor-related protein 5/6 (LRP5/6), thus transmitting extracellular signals to the cytoplasm by phosphorylation of loose protein (Dsh) [91]. (ii) Intracytoplasmic signaling: Dsh prevents β-catenin from phosphorylation or degradation by suppressing GSK-3β activation, thus accumulating free β-catenin [92]. (iii) Intranuclear signal transduction: When the free β-catenin in the cytoplasm reaches a certain level, it can enter the nucleus and combine with the nuclear lymphocyte enhancer factor/T lymphocyte factor (LEF/TCF) to form a β-catenin-LEF/TCF complex, leading to transcription of downstream target genes in the canonical Wnt signaling pathway.

The Wnt/β-catenin system is an evolutionary conserved signaling pathway that is vital for morphogenesis and cell organization during embryogenesis [93]. The expression of Wnt pathway components were up-regulated in hepatic fibrosis using genomic analysis from primary biliary cirrhosis livers [94]. Highly up-regulated expression of Wnt5a and its receptor frizzled 2 (Fz2) implicates this pathway in differentiation of Q-HSCs into myofibroblasts, suggesting an important role of Wnt signaling in development of liver fibrosis [95]. It is not clear whether the role of Wnt5a in promoting fibrosis is caused by inhibition, activation, or independent of β-catenin. A growing number of studies in the literature support the activation of β-catenin by Wnt signaling during HSC activation and fibrosis. Wnt–β-catenin signaling might activate HSCs through negative regulation of adipogenesis, and inhibition of this signaling pathway could contribute to the adipogenic gene profile of Q-HSCs [96]. Inhibiting Wnt signaling to β-catenin therefore might inhibit liver fibrosis. Determining the identity and cell sources of the factors that activate β-catenin in HSCs need further studies. In a single-center, open-label, phase 1 trial, administration of PRI-724, a small-molecule modulator of Wnt signaling, was tolerated by patients with HCV cirrhosis; however, liver injury as a possible related serious adverse event was observed [97].

### 3.4. Hedgehog Signaling

The canonical Hedgehog (Hh) pathway is a conserved, highly complex signaling cascade, with many players and intricate regulation [98]. Patient and mouse data have shown that hepatic fibrosis is associated with Hedgehog activation [99]. It can be simplified into four fundamental components: (i) the ligand Hedgehog, (ii) the receptor Patched (Patch), (iii) the signal transducer Smoothened (Smo), and (iv) the effector transcription factor, Gli. Canonical Hh signaling occurs along a highly specialized organelle, the primary cilium [98]. In the absence of Hedgehog ligand, Patch prevents Smo from entering the primary cilium, repressing Smo activity. This allows the sequential phosphorylation of Gli by several kinases. Phosphorylated Gli is susceptible for ubiquitination by Skip-Cullin-F-box (SCF) protein/β-Transducing repeat Containing Protein (TrCP), which primes Gli to limited degradation in the proteasome. When hedgehog binds to Patch, it removes Patch from the PC, allowing Smo to enter the PC. The entry of Smo into the PC allows Smo activation. Active Smo abrogates phosphorylation and subsequent degradation of Gli. Full length Gli translocates to the nucleus where it acts as a transcription factor for several target genes [100,101].

In healthy adult liver, the Hh pathway expression is relatively dormant with low production of ligands by liver-resident cells and robust expression of Hh inhibitors, such as Hh-interacting protein (Hhip), by Q-HSCs [102]. During fibrogenic liver repair, emerging evidence has demonstrated a critical role of canonical Hh signaling, which supports that conditional deletion of Smo in α-SMA^+^ myofibroblasts inhibited liver fibrosis [103]. Furthermore, Hedgehog ligands can activate HSCs and induce their transdifferentiation from a quiescent phenotype into a myofibroblastic phenotype responsible for matrix deposition [100]. Moreover, the activation of Hh pathway inhibits apoptotic signals, enhances the viability and proliferative capacity of myofibroblasts and stimulates additional production of endogenous Hh ligands in an autocrine or paracrine manner, which drives a positive feedback loop to amplify Hh signaling [104]. Deregulation of the Hh signaling network may contribute to the pathogenesis and sequelae of liver damage [105]. Hhip expression falls by 90%, followed by Shh expression in HSCs and Hh pathway activation [103,106]. During the NIDDK-sponsored PIVENS trial (NCT00063622), treatment response paralleled to loss of Shh^+^ hepatocytes and improvement in Hh-regulated processes that promote NASH progression, indicating that VitE treatment and improvement in NASH were associated with changes in Hh signaling activity [107].

### 3.5. Hippo Signaling

The Hippo signaling pathway is a kinase chain composed of a series of conserved protein kinases and transcription factors, which mainly control organ size by regulating cell proliferation and apoptosis [108]. Hippo signaling is activated by the binding of upstream membrane protein receptors and ligands to generate extracellular growth inhibition signals and activate a group of highly conserved serine/threonine kinases MST [109]. YAP and TAZ are regarded as mechanoactivated coordinators of the matrix-driven feedback loop that intensify and sustains fibrosis [110]. Studies have shown that *TGF*, *PDGF*, *Ankrd1*, *procollagen*, *PAI 1*, *F-actin*, *Fibronectin*, *K19* and other fibrosis-related genes are also regulated by the YAP-TEAD complex [110,111]. Liver injury in mice and humans promotes levels of YAP/TAZ/CYR61 in hepatocytes, hence attracting macrophages to the liver to induce inflammation and fibrosis [112].

YAP accumulates in the nucleus during the early activation of hepatic stellate cells. Inhibiting YAP can prevent HSC activation and fibrogenesis, and reduce the expression of α-SMA and type I collagen [111]. YAP expression was up-regulated in fibrotic liver tissue of model mice induced by CCl_4_ and returned to normal levels after stopping CCl_4_. In HSC-T6 cells treated with TGF-β1, YAP expression increased. In addition, the overexpression of YAP inhibited the apoptosis of activated HSC-T6 cells [111,113]. YAP in the cytoplasm of HSCs enters the nucleus after activation, combines with the transcription factor TEAD1-4, promotes the transcription of genes such as *CTGF* and *PDGF-BB*, and promotes HSC transdifferentiation and proliferation [114]. The I148M variant of the PNPLA3 gene represents a higher risk of severe liver fibrosis, and PNPLA3 I148M up-regulates Hedgehog and YAP Signaling in human HSCs [115]. These indicated that YAP can be used as an effective target to inhibit HSC activation. Conversely, others have argued YAP activation in HSCs is beneficial to liver regeneration. Preventing HSC and YAP activation by manipulating Hedgehog signaling also suppressed liver regeneration and hepatocyte proliferation [116]. Hence, it seems that YAP activation in HSCs represented beneficial non-cell-autonomous effects in the short term but detrimental effects in the long term [117].

During the repair of liver ischemia-reperfusion injury, HSCs were found to be significantly activated and proliferated. LATS1 and its adaptor protein MOB1 (Mps one binder, Mps) are inactivated, and YAP and TAZ in HSCs are selectively activated. At the same time, the expression of CTGF and survivin are up-regulated, and HSC proliferation and concomitant activation of YAP and TAZ occurred not only in injured liver, but not observed in non-ischemic liver. In the process of liver recovery after IR injury, HSC proliferation is obvious [116].

YAP plays a critical role in cell metabolism [118]. The proliferation of activated HSCs exhibits similar metabolic requirements as tumor cells. Studies show essential role of glutamine breakdown in the proliferation and phenotype development of HSCs, which is controlled by Hippo and Hh signaling [119,120].

### 3.6. Crosstalk of Intracellular Pathways

The activation of HSCs is complicated, involving multiple signaling molecules and multiple signaling pathways. These signaling pathways intersect and influence each other, and act together in the entire process of the activation and proliferation of HSCs.

Hippo and TGF-β/SMAD: It is demonstrated that YAP signaling works by promoting the binding of SMAD7 to activated TGF-β receptor type I, thereby eliminating downstream TGF-β signal transduction. At the same time, TAZ binds to SMAD2/3/4 heteromers in a TGF-β-dependent manner and recruits them into TGF-β response elements [121]. *TAZ* knockout experiments also show that TAZ plays a key role in the nuclear accumulation of SMAD2/3/4 complex in response to TGF-β and subsequent transactivation of target genes. In addition, The Hippo pathway scaffold protein RASSF1 is recruited by TGF-β to TGF-β receptor I, and is degraded by the E3 ubiquitin ligase ITCH co-recruited by the receptor, which in turn inactivates the MST/LATS kinase cascade and promotes YAP/SMAD2 interaction and subsequent nuclear translocation [121,122].TGF-β and Notch: Excessive activation of TGF-β regulates the Notch signaling pathway in the process of liver fibrosis in rats. Inhibiting the TGF-β signaling pathway can block the Notch signaling pathway, and Notch signaling can participate in the occurrence of liver fibrosis by activating the TGF-β/SMAD pathway. TGF-β inhibitor down-regulated the expression of Notch1, Hes1 and Hes5, and inhibited Notch signal mRNA and protein expression [123]. TGF-β1 also induced the high expression of Notch1, JAG1, Hes1 in HSC. The expression of the above-mentioned markers in mouse HSC was significantly reduced after *TGF-β1* knockout. After blocking the Notch pathway with specific inhibitors, the expression of Notch1 and α-SMA in HSCs was significantly reduced. These results indicate that TGF-β1 signal controls the activation of HSCs by regulating the expression of Notch signaling pathway markers [124].Hedgehog and Hippo: The activation of the Hedgehog pathway promotes the post-transcriptional response of YAP by increasing the level of YAP protein, so Hedgehog signaling positively regulates YAP [125]. Blocking Hedgehog signaling can inhibit YAP activation in cultured HSCs, and downregulating YAP can inhibit YAP and Hedgehog-induced target gene expression, and inhibit HSC transdifferentiation into myofibroblasts, showing that the Hedgehog pathway can regulate the YAP protein of the regenerated liver in mice [126]. Previous studies have found that the Hedgehog pathway controls the HSCs activation by regulating cellular glycolysis. Conditional interruption of Hh signaling in myofibroblasts reduces the number of glycolytic myofibroblasts and the degree of liver fibrosis in mice [127]. Nevertheless, new research shows that the Hedgehog-YAP signaling pathway can promote the activation of HSCs by regulating the metabolism (i.e., the breakdown of glutamine) during the HSCs transdifferentiation into myofibroblasts [120]. Therefore, glutamine decomposition can control the accumulation of myofibroblasts in mice and may become a therapeutic target for liver fibrosis.Hedgehog and Notch: Activating the Notch pathway in HSCs can stimulate them to become myofibroblasts through a mechanism involving epithelial-mesenchymal transition, which needs to cross the typical Hedgehog pathway. It is suggested that when HSCs are converted to myofibroblasts, it activates Hh signal, undergoes epithelial to mesenchymal transition, and increases the expression of Notch signal. However, blocking Notch signaling in myofibroblasts can inhibit Hh signaling activity and cause mesenchymal epithelial transition; inhibiting Hh pathway can inhibit Notch signaling transduction and also induce mesenchymal epithelial transition [128].

## 4. Conclusions

Chronic liver injury with any etiology can progress to fibrosis and the end-stage diseases cirrhosis and hepatocellular carcinoma. However, currently the development of anti-fibrotic drugs has not yet resulted in clinically approved therapeutics, underscoring the complex biology and challenges involved when targeting the intricate cellular signaling. The current review highlights key extra- and intra-cellular pathways involved in HSC activation, with potential value for the development of refined therapeutic strategies for hepatic fibrosis.

## Figures and Tables

**Figure 1 biomedicines-09-01014-f001:**
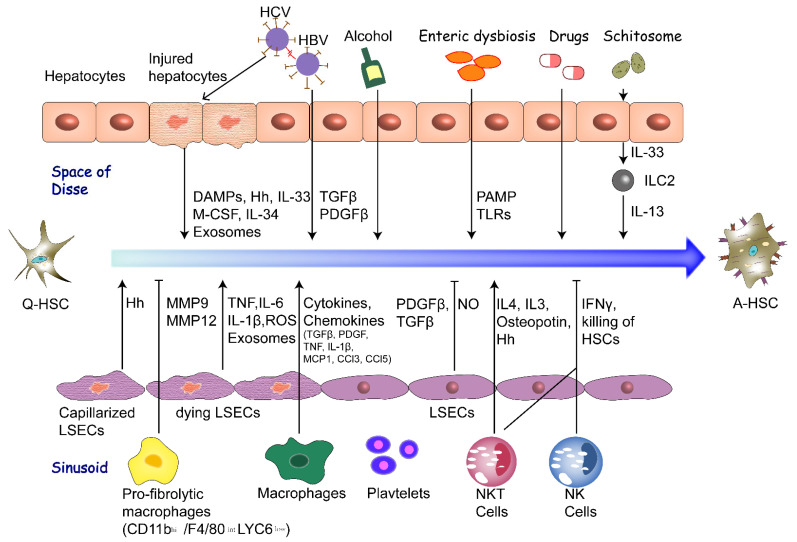
Extracellular factors promoting HSC activation. Extracellular factors including stimulation of various cell types, cyto-kines, altered ECM, hepatitis viral infection, enteric dysbiosis, lipid metabolism disorder, exosomes, alcohol, drugs (MTX, APAP) and schistosome promote or inhibit the activation of HSCs through production of various cytokines and other signaling molecules. The characteristics of A-HSCs include proliferation, contractility, fibrogenesis, inflammatory sig-naling, loss of retinoid and enhanced ECM production. HSC, hepatic stellate cell; DAMPs, damage-associated molecular patterns; Hh, Hedgehog signaling; IL-33, interleukin-33; M-CSF, macrophage colony-stimulating factor; TGFβ, Transforming growth factor β; PDGFβ, Platelet derived growth factor β; PAMP, pathogen-associated molecular patterns; TLRs, Toll-like receptors; MMP, matrix metalloproteinases; TNF, tumor necrosis factor; ROS, reactive oxygen species; MCP1, monocyte chemo-tactic protein-1; NO, nitric oxide; IFNγ, interferon γ; NK, natural killer; NKT, natural killer T; MTX, methotrexate; APAP, acetaminophen; hi, high expression; low, low expression; int, intermediate expression.

**Figure 2 biomedicines-09-01014-f002:**
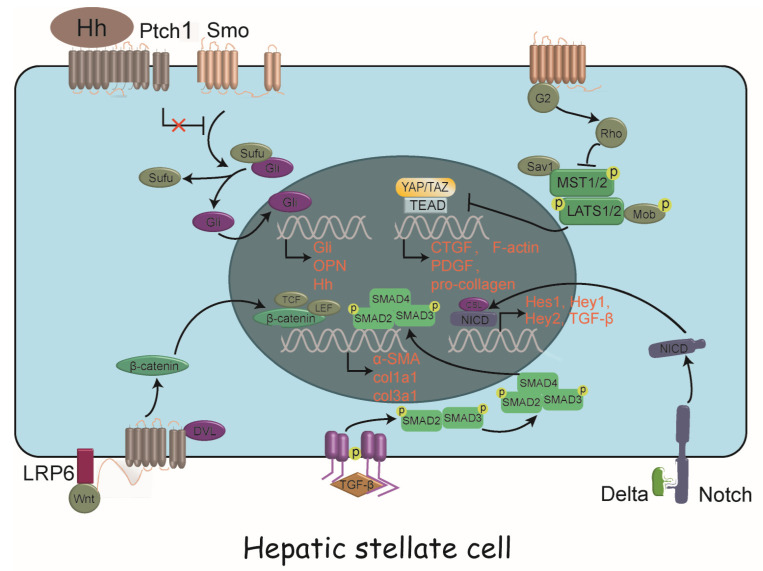
Intracellular signaling pathways driving HSC activation. A panoply of signals drive HSC activation, including TGF-β/SMAD pathway, Notch signaling, Wnt/β-catenin signaling, Hedgehog signaling and Hippo signaling, with complex crosstalk between them. HSC, hepatic stellate cell; Hh, hedgehog; Ptch1, patched 1; Smo, smoothened; Sufu, suppressor of fused; Gli, glioma-associated oncogene homolog; OPN, osteopontin; TCF, T lymphocyte factor; LEF, lymphocyte enhancer factor; LRP6, low-density lipoprotein receptor 6; DVL, disheveled; SMAD, small mother against decapentaplegic; α-SMA, α-smooth muscle actin; col1a1, collagen I α-1; TGF-β,transforming growth factor β; Sav1, Salvador family WW domain containing protein 1; MST1/2, mammalian STE20-like protein kinase 1 and 2; LATS1/2, large tumor suppressor kinase 1 and 2; Mob, monopolar spindle-one-binder protein; YAP, yes-associated protein; TAZ, transcriptional coactivator with PDZ-binding motif; TEAD, TEA domain transcription factor; CTGF, connective tissue growth factor; PDGF, platelet derived growth factor; NICD, Notch intracellular domain; CSL, CBF-1, Suppressor of hairless, Lag-2; Hes, hairy/enhancer of split ; Hey, hairy/enhancer of split related with YRPW motif.

## Data Availability

Not applicable.

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
