# Peer review of "Extra- and Intra-Cellular Mechanisms of Hepatic Stellate Cell Activation"

_biomedicines, 2021, doi:10.3390/biomedicines9081014_

Round 1
Reviewer 1 Report
In the current review, the author discussed the extracellular as well as intracellular mechanism of hepatic stellate cell activation that are associated with the development of hepatic fibrosis.
However, there are few concerns that need to be addressed.
- Extracellular factors responsible for HSC activation are poorly defined throughout and especially hepatocytes, LSECs, altered ECM and enteric dysbiosis section, making it difficult to understand the specific role and mechanism of these factors in HSC activation. The authors need to comprehensively discuss the role of extracellular factors in HSC activation.
- In the section extracellular factors for HSC activation, the authors included viral hepatitis, while there are several other factors that contribute to HSC activation including alcohol and drugs. I understand that it might be difficult to elaborate on all the factors accountable for HSC activation, but the author may at least consider mentioning the name of all the factors associated with HSC activation.
- Under the heading “TGF-β/Smad pathway”, different molecules including Hyaluronan, galectin-3 and others have been discussed abruptly. The author first needs to define the roles of these molecules under normal conditions and then discuss the changes that occur during HSC activation/hepatic fibrosis and how these molecules contribute to the process of HSC activation/ hepatic fibrosis.
- Since this review mainly covers different extracellular and intracellular triggers for HSC activation that derive hepatic fibrosis, the author may consider to include either a figure or a table to summarize the changes occurred in HSCs on molecular/cellular/gene level induced by these factors.
- The section of intracellular signaling pathways of HSC activation is discussed well; however, whether these pathways can be targeted for therapeutic purpose is missing except for TGF-β/Smad pathway. The author may include the relevance of these pathways and discuss whether these pathways have any potential to be targeted for therapeutic purpose.
- Generally, the paper is written well and only minor English editing is required.
Author Response
Dear Editors,
Thank you for giving us the opportunity to resubmit our manuscript biomedicines-1326006 entitled “Extra- and Intra-cellular Mechanisms of Hepatic Stellate Cell”. We have revised the paper in response to reviewers’ comments. The changes are marked in red in the text of the revised version. Please see the attachment.
Yours sincerely,
Changyong Li, MD, PhD.
Department of Physiology, Wuhan University School of Basic Medical Sciences
Email: lichangyong@whu.edu.cn

Reviewer 2 Report
Manuscript number: biomedicines-1326006
Authors: Yufei Yan et al
Title: Extra- and Intra-cellular Mechanisms of Hepatic Stellate Cell Activation
The manuscript by Yufei Yan et al described the mechanisms of Hepatic stellate cell activation. Hepatic stellate cell activation is important issue for understanding hepatic fibrosis. This manuscript is very interesting and well organized. However, some points of criticism arose during reading.
- Recently, it is gaining attention that microRNA and exosome plays important role for hepatic steatosis. The authors are expected to mention about exosome, extracellular vesicles, microRNA.
- In Figure 1, some of the letters are overlapping the lines and are difficult to see, so please correct them.
What is the receptor-like substance on the surface of the A-HSC?
What is the yellow color around the nucleus?
The size of the nucleus of A-HSC is larger than that of s-HSC. Is it reported a difference in the size of the nucleus?
Author Response

(The authors gave the same response as above.)

Round 2
Reviewer 1 Report
As suggested, the authors have made changes in the manuscript.
Reviewer 2 Report
The authors responded to all questions raised, and the manuscript seems to be acceptable in present form.